# Comparative Analysis of Stress in the Periodontal Ligament and Center of Rotation in the Tooth after Orthodontic Treatment Depending on Clear Aligner Thickness—Finite Element Analysis Study

**DOI:** 10.3390/ma14020324

**Published:** 2021-01-09

**Authors:** Jeong-Hee Seo, Emmanuel Eghan-Acquah, Min-Seok Kim, Jeong-Hyeon Lee, Yong-Hoon Jeong, Tae-Gon Jung, Mihee Hong, Won-Hyeon Kim, Bongju Kim, Sung-Jae Lee

**Affiliations:** 1Medical Device R&D Center, DENTIS Co., Ltd., Daegu 41065, Korea; sjh00@dentis.co.kr; 2Department of Biomedical Engineering, Inje University, Gimhae 50834, Korea; eghanacquah@gmail.com (E.E.-A.); kms20162296@gmail.com (M.-S.K.); jeonghyeon6089@gmail.com (J.-H.L.); 3Osong Medical Innovation Foundation, Osong 28160, Korea; yonghoonj186@kbiohealth.kr (Y.-H.J.); bygon@kbiohealth.kr (T.-G.J.); 4Department of Orthodontics, School of Dentistry, Kyungpook National University, Daegu 41940, Korea; mhhong1208@knu.ac.kr; 5Innovative Research & Support Center for Dental Science, Seoul National University Dental Hospital, Seoul 03080, Korea; wonhyun79@gmail.com (W.-H.K.); bjkim016@snu.ac.kr (B.K.); 6Dental Life Science Research Institute, Seoul National University Dental Hospital, Seoul 03080, Korea

**Keywords:** clear aligner, aligner thickness, principal stress, center of rotation, finite element analysis

## Abstract

Lately, in orthodontic treatments, the use of transparent aligners for the correction of malocclusions has become prominent owing to their intrinsic advantages such as esthetics, comfort, and minimal maintenance. Attempts at improving upon this technology by varying various parameters to investigate the effects on treatments have been carried out by several researchers. Here, we aimed to investigate the biomechanical and clinical effects of aligner thickness on stress distributions in the periodontal ligament and changes in the tooth’s center of rotation. Dental finite element models comprising the cortical and cancellous bones, gingiva, teeth, and nonlinear viscoelastic periodontal ligaments were constructed, validated, and used together with aligner finite element models of different aligner thicknesses to achieve the goal of this study. The finite element analyses were conducted to simulate the actual orthodontic aligner treatment process for the correction of malocclusions by generating pre-stresses in the aligner and allowing the aligner stresses to relax to induce tooth movement. The results of the analyses showed that orthodontic treatment in lingual inclination and axial rotation with a 0.75 mm-thick aligner resulted in 6% and 0.03% higher principal stresses in the periodontal ligament than the same treatment using a 0.05 mm-thick aligner, respectively. Again, for both aligner thicknesses, the tooth’s center of rotation moved lingually and towards the root direction in lingual inclination, and diagonally from the long axis of the tooth in axial rotation. Taken together, orthodontic treatment for simple malocclusions using transparent aligners of different thicknesses will produce a similar effect on the principal stresses in the periodontal ligament and similar changes in the tooth’s center of rotation, as well as sufficient tooth movement. These findings provide orthodontists and researchers clinical and biomechanical evidence about the effect of transparent aligner thickness selection and its effect on orthodontic treatment.

## 1. Introduction

Malocclusion refers to uneven teeth or the misalignment of the positions of the maxillary and mandibular dental arches which generally occur due to genetics or acquired factors such as poor dental care and eating habits during infancy [1,2]. Orthodontic treatment involves the use of fixed or removable appliances to correct malocclusion through tooth movements [1,2,3,4].

Fixed appliances consisting of brackets attached to the teeth and an arch-wire connecting them have been commonly used for orthodontic treatment. This orthodontic treatment is, however, of low esthetic satisfaction as they are revealed whenever the patient smiles. Additionally, they cause difficulties in dental care as well as discomfort when chewing [2,4,5]. Eventual enamel wear and damages during debonding post treatment are also of great concern [6,7]. While the emergence of lingual brackets deals with the issues of esthetics, they also pose issues of difficulty in attaching the brackets to the correct teeth location. The use of bonding agents in attaching them also poses dangers of enamel wear or damage [2,6,7]. To resolve these issues, Align Technology, Inc. (San Jose, CA, USA) developed a transparent aligner made of a polymer material that can be attached and detached by the patient and has been used in orthodontic treatments in several patients [2,5,8,9]. These transparent aligners are thought to reduce pain in the facial area due to a lower orthodontic load as compared to fixed appliances. Comparatively, transparent aligners have significantly lower orthodontic forces due to the characteristics of the polymer material. This consequently results in an average tooth movement range of only 0.25 to 0.33 mm. This poses disadvantages of frequent aligner replacements and longer orthodontic treatment periods [1,2,8,9]. While fixed appliances can transmit the orthodontic load to the entire tooth through the arch-wire, transparent aligners are known to only induce limited horizontal movement of the tooth as the orthodontic load is applied only to the exposed crown area [1,2,9]. Due to these limitations of the transparent aligner, they are primarily recommended for the correction of malocclusions in cases that do not require extractions, and where the positions of the central incisors, lateral incisors, and canines, which are relatively easy to correct, are out of the normal dentition range [5,8,9]. Despite their limitations, transparent aligners are currently preferred to fixed appliances by patients in South Korea due to their esthetic properties and reduced treatment pain [10].

The accuracy of orthodontic treatment using transparent aligner treatment is affected by factors such as the thickness of the aligner and attachments to the teeth [5,8,9]. Align Technology recommends establishing the appropriate thickness of the aligner according to the desired amount of tooth movement, and also the shape and position of attachments according to the required amount and direction of tooth movement to shorten the period of orthodontic treatment [11]. The thickness of the aligner is related to the magnitude and maintenance of the orthodontic force applied to the tooth and is considered an important variable in establishing an orthodontic treatment plan considering the condition of the periodontal ligament (PDL) [8,9]. While several studies exist that analyze tooth movement according to the shape and location of attachments, studies investigating the biomechanical analysis of the changes in tooth behavior and periodontal tissue according to the thickness of the transparent aligner remain limited. Gomez et al. [12] and Kim et al. [13] analyzed the biomechanical effect of transparent aligner treatment by implementing a tooth–alveolar bone structure using finite element analysis (FEA). Yokoi et al. [14] analyzed tooth behavior during orthodontic treatment with a transparent aligner using FEA. They performed their analyses for a transparent aligner treatment under the assumption that the alveolar bone was a rigid body, not simulating the actual transparent aligner treatment situation. The above-mentioned studies present a shortcoming as tooth movements and stress analyses in the PDL were performed by directly applying a single load to a single tooth with an attachment, thus not replicating the actual orthodontic loading condition. They also only reported tooth movement and stresses generated in the aligner but not the biomechanical changes in the PDL [12,13,14]. Liu et al. [15] used FEA to compare the stress generated in PDLs according to the thickness of the transparent aligner in a simulation of wearing the transparent aligner. They presented a finite element method that mimicked the actual orthodontic loading conditions of treating malocclusions with transparent aligners. However, their analyses did not indicate the treatment of any malocclusions and only observed the stress response of the PDL when orthodontic forces are applied. The PDL represents a fibrous network connecting the cementum of the tooth root and the alveolar bone. It serves many functions, such as tooth support, nutrition, and protection. During tooth movement, the PDL is pressed in the direction of tooth movement, and tensile behavior is showed in the opposite direction. It is known that the speed of alveolar bone remodeling depends on the biomechanical response of the PDL. It is characterized by nonlinear and viscoelastic properties to relieve the load and impact transmitted from the tooth to the alveolar bone [2,3,16]. However, only a few researchers have applied the nonlinear properties obtained in the PDL test study [17], and most studies do not reflect the actual orthodontic treatment environment by conducting research applying the linear physical properties [15,18,19,20,21].

The aim of this study, therefore, is to present the clinical effect of transparent aligner orthodontic treatment according to aligner thickness by comparatively analyzing the stress distribution in the PDL and changes in the center of rotation of the tooth, using adequately validated dental finite element (FE) models that constitute appropriately represented nonlinear material properties of the PDL.

## 2. Materials and Methods

In this study, a case of Class 1 malocclusion and dentition deformity in the central incisor, which is frequently found in orthodontic transparent aligner treatment, was selected based on previous studies [4,9,22]. Three three-dimensional (3D) dental FE models were constructed to analyze the biomechanical effects of orthodontic treatment of various dentition deformities of a central incisor using FE models of transparent aligners with different thicknesses.

### 2.1. Development of the Intact Dental FE Model

To develop the 3D intact dental FE model, the 3D shape of the mandibular region of a 27-year-old woman with normal occlusion and dentition, who had not received treatment for periodontitis or any other orthodontic treatment, was extracted using MIMICS software (v23.0, Materialize NV, Leuven, Belgium) from cone-beam computed tomography (CBCT) data. Only half of the dentition (half-arch) was used in this study assuming symmetry of the human upper and lower dentitions. This half-arch consisted of the five teeth from the central incisor to the second premolar (tooth number #31 to #35) and excluded the first and second molars because the molars are used as anchorages in the orthodontic treatment for Class 1 malocclusion. The mandibular alveolar bone was separated into cortical and cancellous bones by setting the thickness of the cortical bone at 2 mm. PDLs were created between the teeth and alveolar bone as 0.2 mm offsets using the 3D design program, SolidWorks (v2019, Dassault Systèmes Corp., Vélizy-Villacoublay, France) [13,19,20]. The resulting 3D shapes that constituted the dental FE model were all meshed with four-node tetrahedral elements in ABAQUS (v6.14, Dassault Systèmes Corp.), a commercially available FEA software package (Figure 1).

The tooth–PDL, PDL–alveolar bone, and PDL–gingiva interfaces were assigned “tie” contact interaction conditions. The material properties of the teeth, alveolar bone, gingiva, and PDLs were assigned based on previous studies [13,15,17,18,19,20,21,23,24]. The nonlinear material properties assigned to the PDL in this study are based on the results of the analysis by Natali et al. [24] (Table 1, Figure 2a).

To validate the intact dental FE model developed in this study, a simulation of Periotest, used to measure tooth and dental implant mobility, was performed. A 16N direct hit load in the direction perpendicular to the lingual side of each tooth surface was applied while both sides of the alveolar bone were completely fixed in place. Each tooth’s movement was then measured and compared to the values reported by Parfitt et al. [25].

### 2.2. Implementation of the FE Model for Orthodontic Transparent Aligner Treatment

The transparent aligner model used in this study was shaped and constructed to cover the crown and part of the gingiva. Two of such transparent aligner designs with thicknesses of 0.5 and 0.75 mm were designed. To determine the correct material properties of the transparent aligner to be applied in the FE analysis, material testing was performed using 0.5 and 0.75 mm-thick transparent sheets (Duran CA^®^, SCHEU-DENTAL GmbH, Iserlohn, Germany). These sheets were chosen because they are commonly used by domestic dental appliance manufacturers. Tensile testing was performed for these sheets using a mechanical universal testing machine (MTS Acumen^®^, MTS Systems Corp., Eden Prairie, MN, USA) according to the test standards presented in the South Korean medical device approval guidelines [26]. The stress–strain values from the tensile tests were calculated for each of the aligner sheets (Figure 2b,c) and applied in the FE models.

The lingual inclination malocclusion dental FE model was constructed by rotating the central incisor by 1.5 degrees in the lingual direction through the center of resistance in the sagittal plane. For the axial rotation malocclusion dental FE model, the tooth was rotated 5 degrees in the distomesial direction around the tooth axis. All tooth modifications were set within the allowable range of the transparent aligner treatment suggested by the United Kingdom (UK) National Health Service and Align Technology [11,27]. The orthodontic treatment plan was established by one of the authors, who is an orthodontist, to move the deformed central incisor to the preferred “normal” position (Figure 3).

### 2.3. Loading and Boundary Conditions

Several studies have attempted to perform FE analyses of aligner orthodontic treatment by applying the orthodontic load directly to the tooth or moving the aligner manually [18,19,20,21]. This, however, is not a true representation of the actual orthodontic aligner treatment process. In this study, two separate steps were created in an attempt to mimic the actual orthodontic treatment loading conditions as closely as possible [15]. In the first step, the aligner was affixed onto the FE model of the normal dental dentition and the central incisor was subjected to the desired lingual inclination or axial rotation to generate stresses in the aligner model. The generated aligner stresses were then exported as pre-stresses for the second step. In the second step, the pre-stressed aligners were fixed onto the malocclusion dental FE model. The relaxation of the aligner stresses resulted in correctional tooth movement (Figure 4). Both aligners with different thicknesses were analyzed in the same manner.

In the first step of the analysis, the central incisor was set as a rigid undeformable shell body while the transparent aligner was defined as a deformable body with nonlinear material properties as discussed in Section 2.2. A surface-to-surface frictionless contact condition was assigned to the interface of the aligner and the teeth and the outer base surfaces of the aligner which do not have any contact with the teeth and gingiva were constrained in all directions of movement. To generate pre-stresses in the transparent aligner, translations or rotations opposite to the desired inclination or axial rotation were applied to the central incisor. In the second step, the deformed aligner from the previous step was fixed onto the malocclusion dental FE model. The contact condition between the crown and the transparent aligner as well as the boundary condition for the aligner was set the same as described in step 1. Both sides of the alveolar bone were fixed to prevent movement in all directions. The aligner pre-stresses from the previous step were imported and applied as initial loading conditions for the aligner using the ABAQUS standard user prestress subroutine (Figure 5).

Following the primary analysis of the second step, the step was reiterated for a number of times until the central incisor was adequately corrected to the position of the normal occlusion. The amount of tooth movement was measured and recorded. The principal stresses in the various regions of the PDL were also derived from the ABAQUS program for each tooth correction method and the related aligner thickness. Additionally, changes in the tooth’s center of rotation were also calculated and recorded. Apart from the aligner models, all parts of the dental model remained the same i.e., shape, size, and meshing. Therefore, it was possible to compare and analyze the stress distribution and center of rotation (COR) values for each applied loading for each aligner model using single output values from the FEA program without performing statistical analysis [13,14,15,17,18,19,20,21,28].

## 3. Results

### 3.1. Validation of the Intact Dental FE Model

Periotest was used to validate the intact dental FE model developed in this study. Tooth movement occurring in the horizontal direction in the sagittal plane was measured and compared to the values reported by Parfitt et al. [25]. The FEA-measured values were comparable to those of the experiment by Parfitt et al. Apart from the canine (#33), all FEA measured values were lower than the mean values of the experimental values (Figure 6). All values were within 10 μm of the experimental values. The intact tooth–alveolar bone FE model developed in this study was thus considered to be validated and viable to be used for this study.

### 3.2. Principle Stress in the Periodontal Ligament

In the orthodontic treatment of 1.5° lingual inclination of the central incisor with the 0.5 mm-thick transparent aligner, a buccal lingual inclination of 1.38° was measured. In the case of treating the same malocclusion with a 0.75 mm-thick transparent aligner, a buccal lingual inclination of 1.41° was recorded. The 0.5 mm-thick aligner resulted in 4.77° mesiodistal axial rotation, whereas the 0.75 mm-thick aligner resulted in 4.82° mesiodistal axial rotation during the orthodontic treatment of 5° axial rotation.

The resulting principal stress distributions in the periodontal ligament and central incisor following the treatment for lingual inclination and axial rotation of the central incisor for each aligner model were measured, analyzed, and compared. The results of the analyses showed that principal stress distributions were recorded for the PDL of all teeth (#31 to #35) for both aligner treatments for lingual inclination and are shown in Figure 7a,b. Although in the treatment of axial rotation, principal stress distributions were recorded for all PDLs, the stress values and distributions were predominantly recorded in the central incisor and somewhat the lateral incisor (Figure 7c,d). The values recorded in the PDL of the lateral incisor, canine, and premolars (#32 to #35) were comparatively insignificant to those measured in the central incisor.

In orthodontic treatment for lingual inclination correction, the principal stresses in the lower part of the PDL of the central incisor showed higher values than on the upper part. Even though the lingual side of the PDL received pressure in the middle section, the PDL tended to be tensioned in the lower part. Consequently, compression and tension were measured to be the highest at the lower part of the PDL. The tension in the PDL of the central incisor was measured to be 1.2 MPa in the linguo-mesial (L-M) direction for the 0.5 mm aligner treatment model and 1.6 MPa for the 0.75 mm aligner treatment model. Likewise, the tension values measured in the linguo-distal (L-D) position of the PDL were 1.7 and 2.1 MPa in the 0.5 mm-thick aligner treatment model and 0.75 mm-thick aligner treatment model, respectively. Again, in the buccal (B) position, high compression stresses of −1.5 and −1.8 MPa were measured in the PDL for the 0.5 mm-thick and 0.75 mm-thick aligner treatment models, respectively. Comparatively, the overall induced principal stress in all the regions of the PDL of the 0.75 mm-thick aligner treatment model was higher than in the PDL of the 0.5 mm-thick aligner treatment model. However, the difference was imperceptible (approximately 6% on average) as shown in Figure 8a.

In the orthodontic treatment for axial rotation correction, the principal stresses were concentrated in the middle and upper-middle positions of the PDL of the central incisor. The measured tension values were highest in the mesial position (M) at 0.9 and 1.7 MPa for the 0.5 mm-thick aligner treatment model and 0.75 mm-thick aligner treatment model, respectively. Furthermore, the highest compression values in the PDL were recorded in the labio-mesial (L-M) position at −0.7 MPa for the 0.5 mm-thick aligner treatment model and −0.9 MPa for the 0.75 mm-thick aligner treatment model. In the middle position, a compression value of approximately −0.3 MPa was confirmed for all cases. The difference between the overall induced principal stresses in other areas of the PDL caused by the orthodontic treatment of axial rotation using the 0.5 mm-thick aligner and 0.75 mm-thick aligner was relatively insignificant (approximately 0.03%), as shown in Figure 8b.

### 3.3. Center of Rotation in the Tooth

Table 2 and Figure 9 show a graphical representation of the COR of the central incisor according to results of the first and second analysis steps with respect to the transparent aligner thickness. In the first analysis step where the rigid incisor was inclined or rotated to induce aligner pre-stresses, the COR of the central incisor was located at exactly the same position as the COR indicated on the orthodontic treatment plan for both inclination and axial rotation treatment processes. Conversely, after the second step of the analysis, the COR of the central incisor shifted for all applied displacements. In the orthodontic treatment for lingual inclination, the COR shifted towards the root of the central incisor. For the 0.5 mm transparent aligner model, relative to the COR in the first step, the COR moved 0.8 mm in the lingual direction and 1.3 mm toward the root. For the 0.75 mm transparent aligner model, the COR advanced 0.9 mm lingually and 1.4 mm toward the root, relative to the COR measured in the first step (Figure 9a). For aligner treatment for axial rotation, the COR moved in the labio-distal direction in the crown of the tooth and in the linguo-mesial direction in the root, demonstrating a slanted central axis. With the 0.75 mm transparent aligner, the COR moved chiefly in the labial and distal directions in the crown by 1.2 and 1.5 mm, respectively, relative to the COR measured in the first step. Conversely, in the case of the 0.5 mm-thick aligner, the COR was displaced mostly in the lingual and distal directions in the root by 0.7 and 1.1 mm, respectively, relative to the COR measured in the first step. Especially in the root, the center of rotation tended to have been raised toward the crown, being moved 2 mm higher with the 0.75 mm transparent aligner than that with the 0.5 mm transparent aligner (Figure 9b).

## 4. Discussion

This study aimed to analyze the effect of transparent aligner thickness on the orthodontic treatment of malocclusions by comparing the stress distributions in the periodontal ligaments and the changes in the center of rotation in the tooth using finite element analysis.

The intact dental FE model developed in this study was validated with existing experiments performed on normal subjects [25]. This method of validation was chosen instead of validating the model with other FE models because other FE models are mainly constituted of only linear elastic material properties with no model validation [17]. The overall behavior of tooth response in the current model under a given load was similar to the results of existing experimental studies. Thus, the current model shows potential for application in the study of orthodontic treatment plans and other parameters that are otherwise difficult to visualize experimentally. Liu et al. [15] and Comba et al. [18] suggested that it was possible to study tooth movement even if the PDL is assigned linear material properties. However, in our previous study on tooth movement according to the material properties assigned to the PDL [28], significant differences were observed in the central and lateral incisors and canines. This suggested that dental FE models are sensitive to the material properties assigned to the PDL. Because the biomechanical response of the PDL plays a major role in understanding the principles behind orthodontic treatment mechanisms, the nonlinear properties of the PDL assigned in this model best represent the natural response of the PDL in a finite element model and as such, make this current FE model suitable for analyses related to orthodontic treatments [3,12,21,24].

Generally, the change in the condition of the PDL caused by an external load is considered significant for tooth movement in orthodontic treatment [2,3,24]. Fixed orthodontic appliances consisting of a bracket and an arch-wire apply loads to the tooth by forging the arch-wire shape to produce the suitable amount and direction of tooth movement and connecting it to the bracket attached to the tooth. Owing to this mechanism of action, the tooth is forcibly moved by applying pressure to the PDL and inducing alveolar bone formation and resorption by osteoblasts and osteoclasts between the alveolar bone and the PDL. However, the arch-wire is made of high-strength metal material that transmits excessive loads to the tooth and causes abnormal pressure and tension in the PDL, leading to slow alveolar bone resorption and formation as well as severe pain and discomfort for the patient [1,2,5]. Contrarily, transparent aligners, made of a polymer material, can cause tooth movement by continuously applying a relatively lower load to the tooth, resulting in less pain and discomfort to the patient unlike conventional fixed orthodontic appliances [8,9]. In this study, even though the stress distributions occurring in the PDL of the central incisor following treatment with a 0.75 mm-thick aligner were relatively higher than in the case of 0.5 mm-thick aligner models, the differences were insignificant for both lingual inclination and axial rotation. The principal stress was generally distributed from the middle to the root of the PDL in lingual inclinations and only present in the upper and middle parts of the PDL in axial rotation. The principal stress values recorded after lingual inclination were higher than after axial rotation in general. According to the PDL stress deformation report by Bergomi et al. [3], the stress values measured in this study are well within the range to induce a rapid alteration to the PDL. This also suggests that transparent aligners of 0.5 and 0.75 mm can transmit sufficient pressure to the PDL for orthodontic treatment and correction.

To achieve the desired amount and direction of tooth movement, it is necessary to adjust the load and moment applied to the tooth. The amount and direction of the tooth movement are directly determined by the position of the center of rotation of the tooth [29,30,31]. Smith et al. [29] and Tanne et al. [30] reported that the tooth CR is not fixed but changes depending on the shape and size of the tooth, the condition of the alveolar bone and surrounding periodontal tissue, and the direction of the applied force which can be expressed as the center of movement of the tooth. Smith stated that the center of rotation during tooth movement was located between 1/3 and 1/2 of the long axis of the tooth, whereas Tanne et al. established that the center of rotation was located at 24% of the long axis of the tooth from the root to the crown [29,30]. In an experimental study using artificial teeth–PDL–alveolar bone specimens, Choi et al. [31] reported that the tooth CR moved toward the root of the tooth and was found at a position of approximately 29% from the root to the crown when a vertical load was applied to the tooth surface. The center of rotation reportedly did not move but coincided with the CR in a rotational movement when a simple rotational moment was applied to the tooth. In the present study, during orthodontic treatment using the lingual inclination models, the center of rotation moved towards the lingual and root directions, corresponding to an average of approximately 27.5% from the root of the long axis of the tooth. In the axial rotation models, even though only a simple moment was applied to cause rotational movement of the tooth as suggested by Choi et al., the center of rotation largely moved diagonally away from the long axis of the tooth. This can be attributed to the size of the tooth and the height of the alveolar bone as suggested by Tanne et al. [30]. Additional studies about the shifting of the center of rotation during tooth movement when a simple movement is applied are needed. Furthermore, when planning orthodontic treatment, the position of the center of rotation should be established with respect to not only the abstract CR but also the depth of the tooth in the alveolar bone and the thickness of the surrounding periodontal tissue. The results of these studies are expected to be helpful in the analysis of the changes in the center of rotation for orthodontic treatment using transparent aligners. In this study, the thicker transparent aligner resulted in a further shift of the center of rotation of the tooth away from the abstract CR, transmitting more load to the tooth than the thinner transparent aligner. Consequently, the goal of the orthodontic treatment was reached with fewer iterations using the thicker transparent aligner than the thinner aligner. A more detailed analysis is however required to confirm that the tooth was indeed displaced to the prescribed final position.

To achieve orthodontic treatment with transparent aligners, pre-stresses were successfully generated in the transparent aligner and applied as loading conditions for tooth displacements. By using this method of finite element analysis, the orthodontic treatment with transparent aligners was adequately simulated without having to omit the alveolar bone, which plays a major role in orthodontic treatment, or directly applying loads on the tooth surface without the transparent aligner. This study only focused on the orthodontic treatment for malocclusion using only transparent aligners. Therefore, comparative analyses of the efficiency of orthodontic treatment using transparent aligner and orthodontic treatment using conventional options like brackets and arch-wire combinations could not be established. Additionally, the effect of attachments that are sometimes affixed to the teeth during aligner treatments was not considered. Furthermore, this study only examined cases of unidirectional and non-severe deformations in one half of the dental arch with some teeth that are eligible for transparent alignment treatment. In future research works, orthodontic treatment of not only non-severe unidirectional deformation but also complex severe deformations requiring loading conditions in two or more directions need to be considered. In addition, other forms of orthodontic treatment options need to be analyzed to be able to biomechanically evaluate the detailed advantages and disadvantages of interventions for malocclusion using transparent aligners and other existing orthodontic treatments in a full dental arch.

## 5. Conclusions

This study compared and analyzed the stress distributions in the periodontal ligament of the central incisor and the center of rotation in the central incisor according to the thickness of the transparent aligner during orthodontic treatment for lingual inclination and axial rotation using finite element analysis. The principal stresses induced by the aligner in the periodontal ligament of the central incisor were within ranges that were sufficient to induce the remodeling of the periodontal ligament to produce sufficient tooth movement. The 0.75 mm-thick aligner resulted in a slightly higher orthodontic load on the tooth than the 0.5 mm-thick aligner as the tooth center of rotation shifted from the abstract center of rotation.

## Figures and Tables

**Figure 1 materials-14-00324-f001:**
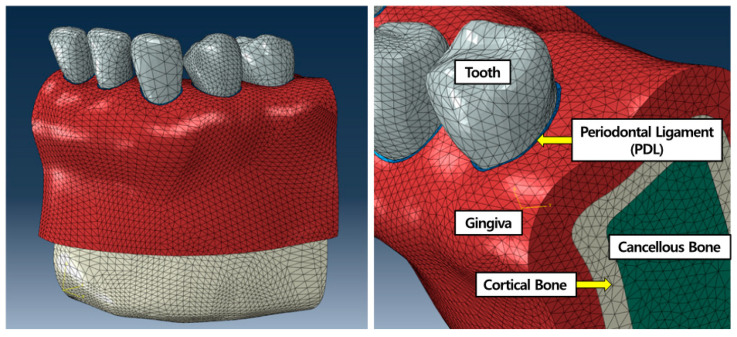
Intact dental finite element (FE) model showing the teeth, gingiva, PDLs, and cortical and cancellous bones.

**Figure 2 materials-14-00324-f002:**
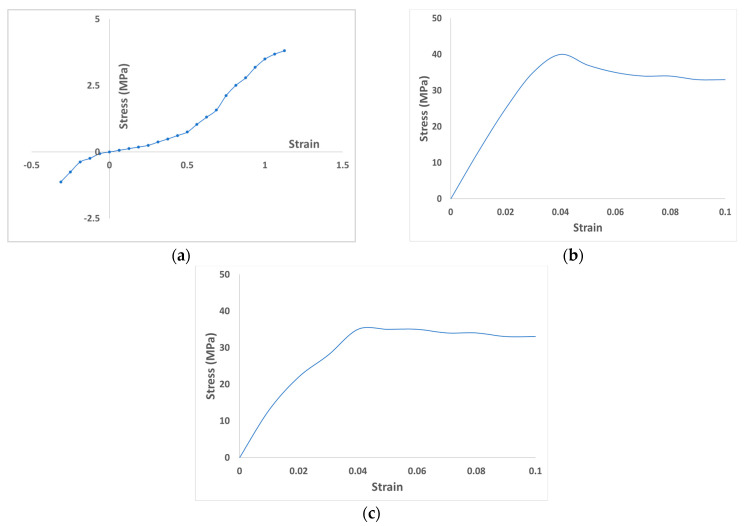
Stress–strain curve for nonlinear material properties of (**a**) PDL, (**b**) 0.5 mm-thick aligner, and (**c**) 0.75 mm-thick aligner.

**Figure 3 materials-14-00324-f003:**
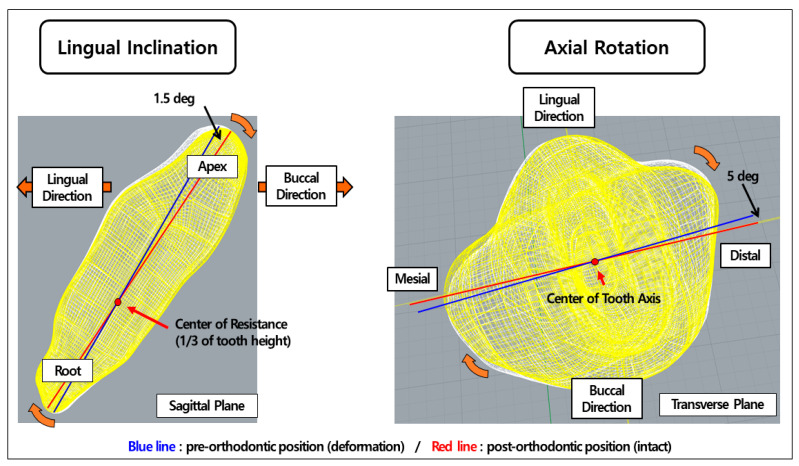
Deformation types and orthodontic treatment plans of central incisor applied to finite element analysis (FEA).

**Figure 4 materials-14-00324-f004:**
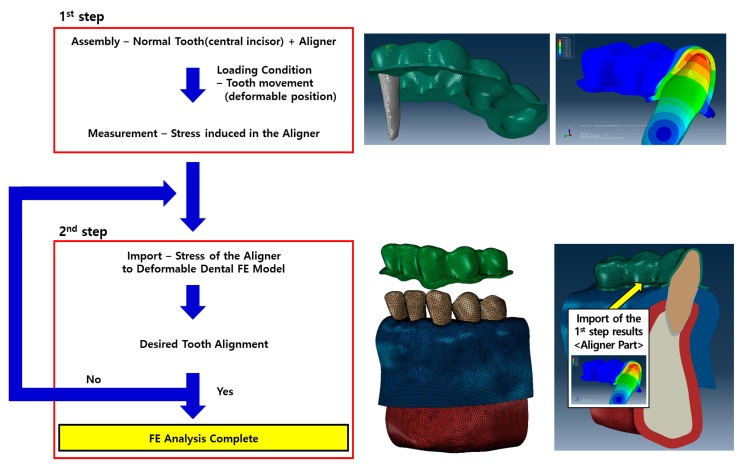
Process of the FEA in orthodontic treatment using the clear aligner.

**Figure 5 materials-14-00324-f005:**
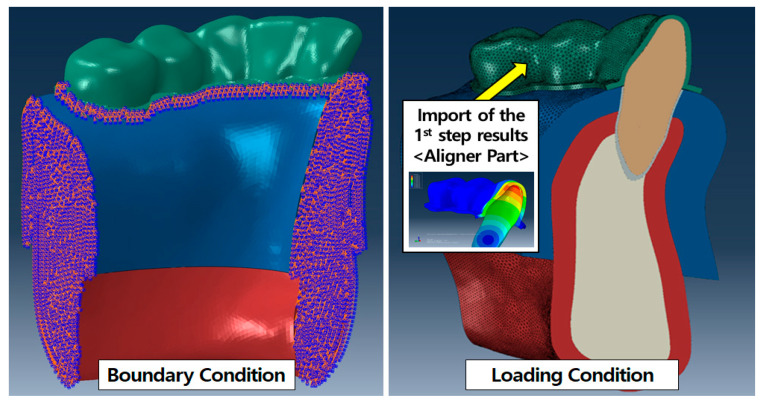
Loading and boundary condition of the 2nd step in FEA.

**Figure 6 materials-14-00324-f006:**
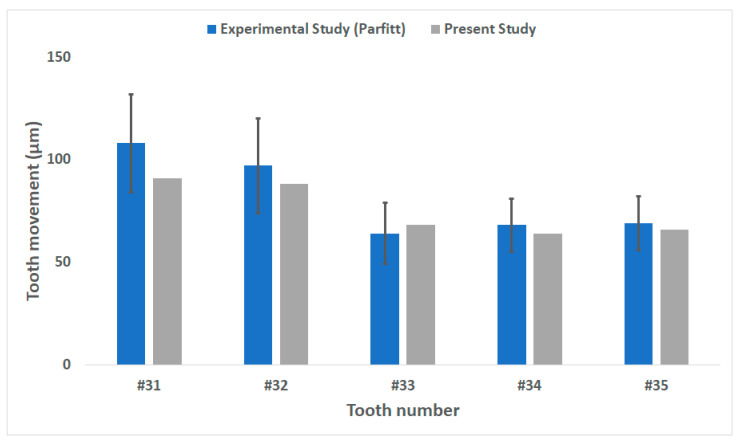
Validation of the intact dental FE model with tooth mobility analysis measurements.

**Figure 7 materials-14-00324-f007:**
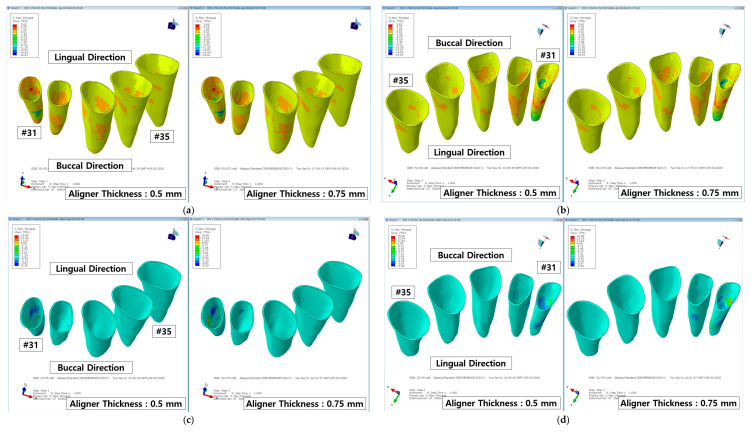
Principal stress distributions in the PDLs following transparent aligner orthodontic treatment of the central incisor. (**a**) buccal- and (**b**) lingual-direction view after lingual inclination correction; (**c**) buccal- and (**d**) lingual-direction view after axial rotation correction.

**Figure 8 materials-14-00324-f008:**
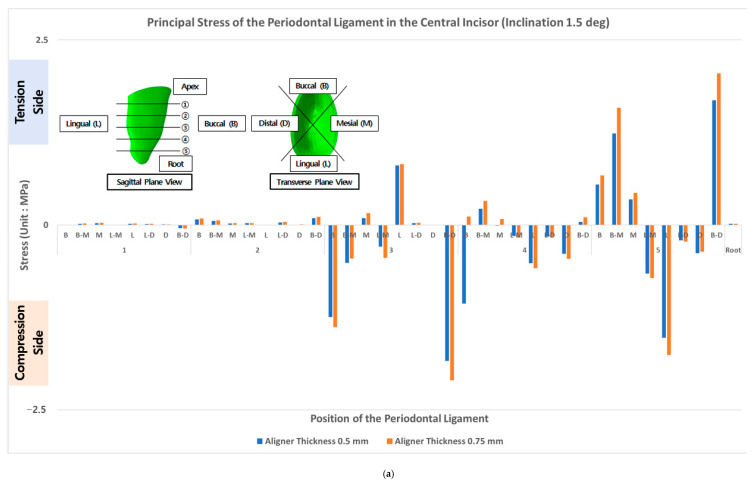
Principal stress in the PDL of the central incisor following aligner treatment procedure. (**a**) Lingual inclination and (**b**) axial rotation correction.

**Figure 9 materials-14-00324-f009:**
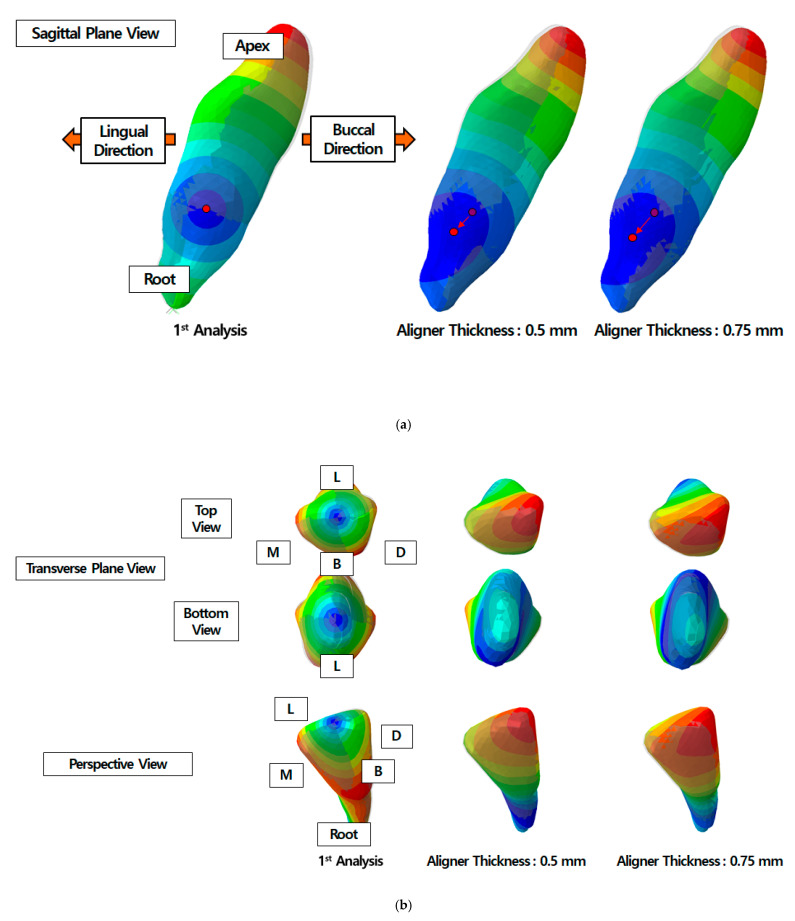
Change in the center of rotation for the central incisor. (**a**) Lingual inclination and (**b**) axial rotation of the central incisor deformation. (L: lingual, B: buccal, M: mesial, and D: distal direction).

**Table 1 materials-14-00324-t001:** Material properties of the components in the FE models—linear parts.

Component		Elastic Modulus (MPa)	Poisson’s Ratio
Alveolar bone	Cortical bone [13,18,19]	13,700	0.3
	Cancellous bone [13,18,19]	1370	0.3
Teeth [13,20,21]		19,613	0.15
Gingiva [13,23]		2.8	0.4

**Table 2 materials-14-00324-t002:** The location change of the center of rotation (COR) values (mm) after orthodontic finite element analysis (FEA) in two orthodontic treatment situations. (Origin position: COR in orthodontic treatment planning).

Orthodontic Treatment	Direction		Distance of the COR	
			0.5 mm-Thickness	0.75 mm-Thickness
Inclination	Apex (+)-Root (−)		−1.3	−1.4
	Buccal (+)-Lingual (−)		−0.8	−0.9
Axis Rotation	Apex	Mesial (+)-Distal (−)	−0.2	−1.4
		Buccal (+)-Lingual (−)	+0.3	+1.8
	Root	Mesial (+)-Distal (−)	+0.1	−1.0
		Buccal (+)-Lingual (−)	−0.1	−0.8

## Data Availability

Data sharing not applicable.

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
