# Peer review of "Comparative Analysis of Stress in the Periodontal Ligament and Center of Rotation in the Tooth after Orthodontic Treatment Depending on Clear Aligner Thickness—Finite Element Analysis Study"

_materials, 2021, doi:10.3390/ma14020324_

Round 1

Reviewer 1 Report

the study is well conducted, but there are some information to be added in the intro and eventually discussed in the discussion. 

Indeed other disadvatages of the fixed orthodontics is the resuals of resins and eventual enamel damages in the debonding and clean up procedures 

https://pubmed.ncbi.nlm.nih.gov/32560482/

even of smaller and more aesthetics, align technology include the placement of "brackets". This part should be mentioned and discussed. 

limit is the type of considered tooth...what about the loading in other teeth?

Reviewer 2 Report

The present study has demonstrated the effect of Clear Aligner, which is used in 0.5mm and 0.75mm thickness, on periodontal ligament and center of rotation in the tooth during the orthodontic treatment through finite element analysis method. This paper has practical values, however there exists some expository short comings as detailed in the follows.

  1. The experimental date in the result section (such as tooth movement, stress of the periodontal ligament) has lacked statistical analysis to determine whether there is insignificant difference, and statistical analysis method should be added in the section of materials and method.
  2. It would be preferred to add a table for quantitative analysis of the movement of the center of rotation in the section of result to make the article more readable.
  3. The clarity of Figure 7 and Figure 8 should be improved.
  4. Please added the corresponding reference to statement in line 86-89.

Reviewer 3 Report

The manuscript submitted to Materials entitled “Comparative Analysis of Stress in the Periodontal Ligament and Center of Rotation in the Tooth after Orthodontic Treatment depending on Clear Aligner Thickness – Finite Element Analysis Study” is an original article which aim to investigate the effect of orthodontic treatment by comparing the stress in the periodontal ligament(PDL) and the center of rotation in the tooth according to the thickness of the clear aligner developed to treat malocclusion through finite element analysis(FEA).

On my opinion the article is interesting, well written, with good English. The content of the manuscript is interesting.

However, I highlighted some issues.

Abstract. It may be helpful to structure the abstract to attract the reader's attention.

Introduction. Specify if there are other similar studies. Better specify the objectives and methods of the study.

Please remove all these sentences at lines 87-91 on page 2:

<< During tooth movement, the PDL is pressed in the direction of tooth movement, and tensile behavior is showed in the opposite direction. It is known that the speed of the alveolar bone remodeling depends on the biomechanical response of the PDL. The PDL is a soft tissue located between the tooth and the alveolar bone, having nonlinear and visco-elastic properties to relieve the load and impact transmitted from the tooth to the alveolar bone [2,3].>>

Insert the following sentence at line 87 on page 2:

<< The PDL represents a fibrous network connecting the cementum of the tooth root and the alveolar bone. It serves many functions, such as tooth support, nutrition, and protection. During tooth movement, the PDL is pressed in the direction of tooth movement, and tensile behavior is showed in the opposite direction. It is known that the speed of the alveolar bone remodeling depends on the biomechanical response of the PDL. It is characterized by nonlinear and visco-elastic properties to relieve the load and impact transmitted from the tooth to the alveolar bone [2,3, https://doi.org/10.1177/0963689718807680].>>

Discussion. Are there other similar studies that have shown similar results? Did the authors find limitations in their study by comparing it with other in the literature?

Figures. Improve the quality and arrangement of figures.

After making the indicated changes, I am available for a second round of peer review.

Round 2

Reviewer 3 Report

The article is now suitable for publication.